# Normal Range for the Schirmer Tear Test and Intraocular Pressure in Healthy Latvian Darkhead Lambs and Ewes

**DOI:** 10.3390/vetsci10060392

**Published:** 2023-06-08

**Authors:** Liga Kovalcuka, Nelli Alexandra Margarethe König, Pia Valentina Helen Petersen, Aija Sneidere, Aija Malniece

**Affiliations:** Clinical Institute, Faculty of Veterinary Medicine, Latvia University of Life Sciences and Technologies, LV-3004 Jelgava, Latvia

**Keywords:** intraocular pressure, Schirmer tear test, sheep, lamb, ewe

## Abstract

**Simple Summary:**

Sheep are susceptible to various ophthalmic conditions, such as congenital, acquired, and infectious diseases. To obtain an accurate diagnosis and avoid misinterpretation, a comprehensive ophthalmological examination that includes diagnostic tests, such as the Schirmer tear test (STT) and intraocular pressure (IOP) measurement in particular species and breeds, is essential. Decreased STT can illustrate dry eye syndrome (*keratoconjunctivitis sicca*), but increased STT can characterize ocular discomfort and pain. Tonometry, at the same time, is useful to diagnose intraocular inflammation—uveitis. Normal STT and IOP values have been described in the literature for small animals such as cats and dogs; however, they are poorly described for small ruminants such as sheep. As far as we know, there are no reports of IOP values in lambs and no reports on STT and tonometry in Latvian Darkhead sheep. Therefore, this study aimed to improve data and determine the reference range for STT and tonometry values in clinically normal Latvian Darkhead lambs and ewes and will provide specific diagnostic values for Latvian Darkhead sheep.

**Abstract:**

A complete ophthalmological examination, including basic diagnostic tests such as the Schirmer tear test (STT) and intraocular pressure (IOP) measurement performed by tonometry in particular species and breeds, is essential for obtaining a clear diagnostic result and avoiding diagnostic misinterpretations. STT and IOP values have been poorly described for sheep. Therefore, this study aimed to determine the normal range for STT and tonometry values in clinically normal Latvian Darkhead lambs and ewes. Both eyes of 100 sheep (200 eyes)—50 lambs (1–3 months old) and 50 ewes (1–8 years old)—underwent complete ophthalmic examinations, including STT and IOP evaluation. The mean ± standard deviation of STT values for both eyes in lambs and ewes were 13.12 ± 3.91 mm/min and 13.68 ± 4.09 mm/min, respectively. The IOP in lambs and ewes was 14.04 ± 3.68 mmHg and 19.16 ± 3.24 mmHg, respectively. Furthermore, the suggested reference range for the STT was 12.00–14.23 mm/min in lambs and 12.52–14.84 mm/min in ewes, while the reference IOP range was determined to be 13.00–15.08 mmHg in lambs and 18.24–20.08 mmHg in ewes. No statistically significant differences in STT and IOP values were observed for both eyes. However, the IOP value for both eyes was statistically significantly higher in ewes compared to lambs (*p* < 0.01). This study provides reference values for the STT and IOP in healthy Latvian Darkhead lambs and ewes.

## 1. Introduction

Sheep are susceptible to various ophthalmic conditions, such as congenital, acquired, and infectious diseases. During a complete clinical ophthalmological examination, measurement of relevant ocular parameters through basic diagnostic tests, such as the Schirmer tear test (STT) and tonometry for evaluation of intraocular pressure (IOP), is essential to ensure adequate diagnosis [1]. The STT is essential to evaluate physiological tear production volume in animals and involves quantitative measurement of aqueous tear amounts. The precorneal tear film covers the eye’s surface and the aqueous portion, located at the middle layer of the tear film, is important for maintaining ocular surface health in all animal species. Tears are also essential for removing foreign matter such as dust and plant parts, for providing nutrients and oxygen to the avascular cornea and conjunctiva, and for providing corneal defense via immunoglobulins, lysozymes, and other proteins [1,2]. Decreased tear production most commonly can cause keratoconjunctivitis sicca or dry eye syndrome, leading to ocular discomfort, pain, and in more severe cases to corneal and conjunctival inflammation, corneal epithelial or stromal ulcers, and even perforation of the cornea and vision loss. Ocular discomfort and pain can affect animal welfare and normal weight gain. In comparison, an increase in STT values can be a significant ophthalmic sign of an ocular foreign body, eye irritation, pain, inflammation, or nasolacrimal system dysfunction [1,3,4].

Intraocular pressure results from the balance between aqueous humor production and its outflow in the eye [1,2], and can be measured using a tonometer, such as a rebound tonometer (TonoVet^®^, Tiolat Ltd., Vantaa, Finland). Tonometry is essential for diagnosing low-pressure conditions in the eye, such as uveitis, eye penetration, and perforation, and high-pressure conditions, such as glaucoma. Both conditions are painful and can lead to vision loss in animals [1,2,5].

Normal STT and IOP values have been described in the literature, mainly for small animals such as cats and dogs; however, they are poorly described for small ruminants such as sheep [2,6,7,8,9]. STT values have been described in several sheep breeds, including Merinos, Barbary, Lacoma, Chios, Florina, and Lacaune mixed breeds [10,11,12]. However, only one report of STT values in lambs has been found [10]. Additionally, there are reports of IOP in Corriedale and Awassi sheep measured using the Tonopen-XL and Perkins tonometers [13,14], and only a few reports on IOP measured using the rebound TonoVet^®^ tonometer [9]. Based on a thorough literature search, using the keywords “IOP”, “intraocular pressure”, “tonometry”, “lambs”, “sheep”, “ewe”, “TonoVet”, there are no reports of IOP values using TonoVet^®^ in lambs. Therefore, it is important to establish normal values of STT and IOP for lambs and ewes, especially for specific regional breeds such as Latvian Darkhead sheep.

The study aimed to determine the normal range for tear production using the standard STT and IOP measured by using the rebound TonoVet^®^ tonometer in Latvian Darkhead lambs and ewes.

The study results will improve data on normal diagnostic values for lambs and ewes and will simultaneously show specific diagnostic values for Latvian Darkhead sheep.

## 2. Materials and Methods

This study was performed in full conformance with the ethical criteria and welfare of the sheep involved. All examined animals were privately owned Latvian Darkhead sheep from a farm in Vidzeme, Latvia (57.45, 24.83), and informed consent was obtained from the farm owner. The examination procedures were performed during routine clinical and ophthalmological examinations at the time of the planned deworming treatment. The procedures did not exceed the principles of good veterinary practice or animal welfare and were not painful. This study was approved by the Ethical Commission of Latvia University of Life Sciences and Technologies Faculty of Veterinary Medicine (Nr. LLU_Dzaep_2022-2-4). The Animal Ethics Board waived the need for permission for animal ethics in the general Latvian Republic Food and Veterinary Service.

The study was performed on two separate occasions: 5 June 2021 and 17 June 2022, approximately during the same time of the day (10 am–4 pm).

This study involved 100 Latvian Darkhead sheep (200 eyes), including 50 healthy lambs (1–3 months old; 21 males and 19 females), 100 eyes, and 50 adult sheep (50 ewes; 1–8 years old), 100 eyes, respectively.

In order to avoid the influence of subclinical sheep diseases on the results of this study, a Latvian Darkhead sheep pedigree flock with Maedi Visna (MV) free herd status M3 from 2014 and Brucellosis free status from the year 2010, according to the official Maedi Visna and Brucellosis control programs stated by the Republic of Latvia, was selected. 

All the sheep included in this study underwent a complete ophthalmological examination of both eyes performed by the same person to ensure that they were healthy. General clinical examination included animal signalment (animal breed, age, and sex), general appearance, vital signs (temperature, respiratory rate, and heart/pulse rate), and physical examination using a systems approach. To ensure a good animal health status for animals used in this study, body condition score, mucous membrane color, and fecal staining of the perineal region while going through the sheep handling system data were assessed. Only animals with body condition scores of 2.5–3.5 points, pink mucous membranes, and clean perineal regions were further passed through the scale system for ophthalmological examination and data obtaining for research purposes. The above-mentioned examination minimized the possibility of serious or noteworthy health disorders such as heavy endoparasites or paratuberculosis presence.

Ewes and lambs were randomly selected by sending all animals through farm electronic scales, every third animal was selected, and an ophthalmic examination was performed. Ophthalmic examination included (i) basic neurological and vision tests (menace and dazzle test and direct and indirect pupillary light reflex). Both eyes were examined with (ii) slit-lamp biomicroscopy (Kowa SL15; Nagoya, Aichi, Japan) and monocular ophthalmoscopy with a PanOptic ophthalmoscope (Welch Alynn, Romford, UK), without topical mydriatic agent application. Tonometry with a TonoVet^®^ tonometer (TonoVet^®^, Tiolat Ltd., Finland) and tear production evaluation using standardized sterile STT strips (Eickemeyer, Tuttlingen, Germany) were performed in both eyes.

Animals with epiphora, lacrimation, eye discharge, blepharospasms, or any other signs of clinical ophthalmic or systemic disease were excluded from the research, and the next animal was selected for testing. Each animal was individually examined in a restraining box. The number on each animal’s ear tag was recorded, then the animal’s head was restrained, and the STT and tonometry were performed. During measurements, the animals were gently handled to avoid any tension on the animal neck, which might have influenced the IOP. Afterwards, the animals were weighed and released.

A standardized STT-I was performed by inserting the end of the sterile test strip inside the lower lateral eyelid margin in the conjunctival fornix for 60 s (1 min). After removing the test strip, the length of the wet part of the strip was immediately measured in millimeters.

Tonometry was performed by the same person, and IOP measurements were obtained using a rebound tonometer (TonoVet^®^, Tiolat Ltd., Finland) calibrated on the (d) calibration setting for use in dogs, as provided by the manufacturer. From five successive IOP readings, the average for this particular tonometer was obtained. Topical anesthesia is not required when using this tonometer, which benefits animals, because some authors have reported that corneal endothelial and systemic toxicity can occur with the frequent use of topical anesthetics [15,16]. A single-use probe was positioned perpendicular to the corneal surface, approximately 4–6 mm from the central cornea. During measurement, care was taken to prevent compression of the animal’s jugular veins or cervical region.

All animals during both IOP and STT assessment episodes were located on pasture, and no additional concentrate feed was supplied for ewes or lambs, thereby minimizing the possibility of subclinical ruminal acidosis effects on animals’ health status.

All measurements were performed during the daytime (10.00 am–4.00 pm) to minimize the effects of changing light conditions at different times of day on the IOP [6,17,18].

The sample size was calculated with automatic power analysis in the program G-power. The number of 45 animals per group was set according to the G-power calculation at an effect size of 0.6, a Type I error (α) of 0.05, and a Type II error (power, β) of 0.80. We decided to work with a sample size of 50 ewes and 50 lambs, using 100 animals in total, and to obtain 200 measurements of IOP and STT values.

Statistical analysis was performed using Statistical Product and Service Solutions (SPSS, version 12.0.0, SPSS Inc., Chicago, IL, USA and Microsoft Office, Excel, version 2016, Microsoft Corp., Redmond, WA, USA). The arithmetic mean values (X) are presented as the mean ± standard deviation (SD). The STT and IOP reference values were evaluated for each eye separately and for both eyes together in lambs and ewes.

Normality was tested using the Shapiro–Wilk and Wilcoxon Signed Ranks tests. A paired sample *t*-test was used to compare the STT, IOP obtained from the right and left eyes, and differences between sex and age. Additionally, *p*-values < 0.05 were considered statistically significant.

## 3. Results

During and after the study, no signs of ocular irritation or pain were detected in any ewes or lambs at any time point during the study.

### 3.1. Schirmer Tear Tests in Lambs and Ewes

The STT was performed for all 50 Latvian Darkhead lambs (100 eyes), and the STT ± SD values of the right and left eyes in lambs and ewes are presented in Table 1. No statistically significant differences were observed between the right and left eye (*p* > 0.05), showing STT values in lambs of 13.12 ± 3.92 mm/min in the right eye and 13.22 ± 3.94 mm/min in the left eye. The mean STT ± SD values of both eyes were 13.12 ± 3.91 mm/min, and the suggested reference range of the STT was determined to be 12.00–14.23 mm/min. The STT measurements for lambs showed minimum values from 7 mm/min to a maximum of 23 mm/min, with a median of 12.50 mm/min in both eyes (Table 1).The mean STT ± SD value in 50 Latvian Darkhead ewes was 13.68 ± 4.09 mm/min in both eyes, with values of 12.52–14.84 mm/min in both eyes, respectively. The minimum values were 5 mm/min to a maximum of 24 mm/min, with a median of 13 mm/min in both eyes. No statistically significant differences were observed between the STT values of the right and left eyes in ewes (*p* > 0.05), showing STT values of 13.68 ± 4.09 mm/min in the right eye and 14.58 ± 4.59 mm/min in the left eye, respectively. Descriptive statistics and range intervals for the STT values in lambs and ewes are summarized in Table 1. No statistically significant differences were observed in the STT values of the lambs and ewes (*p* > 0.05).

### 3.2. Intraocular Pressure in Lambs and Ewes

The mean IOP ± SD of the right eye were 14.04 ± 3.68 mmHg and 14.06 ± 4.38 mmHg in the left eye, and both eyes in Latvian Darkhead lambs were 14.04 ± 3.68 mmHg, with suggested reference range of 13.00–15.08 mmHg, and no statistically significant differences in IOP were observed between the eyes (*p* > 0.05). The descriptive IOP values of the right and left eyes of lambs and ewes are shown in Table 2. The IOP measurements for lambs showed minimum values from 9 mmHg to a maximum of 22 mmHg, with a median of 14 mmHg in both eyes (Table 2).The mean IOP ± SD of the right eye was 19.16 ± 3.24 mmHg and the left eye 19.28 ± 3.18 mmHg, both eyes in ewes were 19.16 ± 3.24 mmHg, with values of 18.24–20.08 mmHg (Table 2). Additionally, no statistically significant differences in IOP values were observed between ewes’ right and left eyes. The IOP values were statistically significantly higher in both eyes in ewes than in lambs (*p* < 0.01). The tonometry for ewes showed minimum values from 10 mmHg to a maximum of 24 mmHg, with a median of 19 mmHg in both eyes. The descriptive statistics and the values for IOP in lambs and ewes are summarized in Table 2.

## 4. Discussion

Normal Schirmer tear test values are crucial for a complete clinical and ophthalmological examination of any animal. Furthermore, knowing the reference values for specific species and breeds can lead to clear diagnostic results and prevent diagnostic misinterpretations during ophthalmic examinations. To the best of our knowledge, following a thorough literature search process, this is the first study conducted on the Latvian Darkhead sheep breed, but more importantly, in lambs. This study found a mean STT value in lambs of 13.12 ± 3.81 mm/min in both eyes, and the suggested reference range for STT in Latvian Darkhead lambs was 12.00–14.23 mm/min. Isler et al. (2013) showed higher STT values in Merinos lambs at 17.33 ± 1.54 mm/min in females and 16.06 ± 1.52 mm/min in male lambs in both eyes, respectively. However, the age of lambs included in their study was significantly different (15–20 days) from the age (1–3 months) of lambs in the present study [10]. In addition, Akgül et al. (2022) compared the STT values of 1–6 days old healthy lambs to lambs with *E. coli* infection and found that the results were significantly lower for the lambs infected with *E. coli*. They reported the average values of STT for the healthy lambs to be 16.6 ± 8.54 mm/min in both eyes, higher than in our study [19].

Several studies have determined the average STT values in healthy ewes of different breeds such as Merino, Chios, Florina, and Lacaune, and in mixed breeds and geographic locations such as Turkey and Greece [4,10,19]. However, Dedousi et al. (2019) showed that tear secretion (STT range 10.8–26.2 mm/min) was similar regardless of breed when examined under the same conditions [4]. In our study, all measurements were performed under the same conditions, the same period of the year, and the same breed; however, as a limitation of our study, we assume that still some variations are possible between different farms and farming conditions. Seasonal effects have also been reported; STT values were found to be significantly higher in summer compared to wintertime [4]. Latvia is a typical Northern European country, where the average temperatures during summer and winter differ from those in other Southern European countries. For example, the average temperature in Latvia during the Summer in June is reported to be +17 °C and on some occasions can reach +26 till +32 °C, and in Greece and Turkey it is reported to be +25 °C and +19 °C, respectively [20]. Therefore, obtaining the STT values in this particular region is important. To the authors knowledge this is the first study of evaluating STT values conducted on farm animals in the Northern European region. For the future, STT values should also be determined during the winter period. 

A study including 90 ewes of three different breeds, such as Chios, Florina, and Lacaune, reported average STT values of 18.45 ± 3.93 mm/min in both eyes, and the reference range for summer was 13.8–26.7 mm/min [4]. These values are higher than those reported in the present study (13.68 ± 4.09 mm/min). However, the reference range for summer reported in their study overlaps partly with the values observed in the present study. Similarly, there are more reports on Romanov and Ghezel sheep, where the STT reference values vary between 16.7 ± 4.9 mm/min and 19.0 ± 5.22 mm/min in both eyes, respectively, showing higher STT results than those in Latvian Darkhead sheep [21,22]. Furthermore, Kurt et al. (2021) recently determined the mean STT values for Sakiz sheep to be 11.8 ± 3.35 mm/min [8], which is slightly less than those for the Latvian Darkhead breed. Therefore, it is important to establish a reference range for the STT in a particular breed in different geographic locations.

In our study, no significant differences were observed in the STT values between lambs and ewes.

The current study also aimed to establish reference values for intraocular pressure in clinically normal Latvian Darkhead sheep by using a rebound tonometer (TonoVet^®^). In Latvian Darkhead lambs, the reference range of IOP was calculated as 13.00–15.08 mmHg (mean ± SD: 14.04 ± 3.68 mmHg in both eyes). In ewes, the normal range of IOP was 18.24–20.08 mmHg (mean ± SD: 19.16 ± 3.24).

To the best of our knowledge, only one study has measured IOP in lambs; younger than in our study, only 15–20 days old Merino lambs showed slightly higher IOP values of 16.46 ± 4.25 mmHg [10], but in this study, the Schiotz tonometer was used. We need to admit that the Schiotz tonometer is an impression (invaginating) tonometer, and it is not a commonly used tonometer in veterinary practice anymore, and its values are not directly comparable. At the same time, the performance of this tonometer determines a tighter fixation and lifting of the sheep head, thus increasing a protentional IOP values. Therefore, we can confirm that this is the first study to evaluate IOP in sheep lambs by using a rebound tonometer TonoVet^®^.

Several studies have investigated the average IOP values in sheep, especially in ewes, but also with different tonometers and sheep breeds such as Sakiz, Merino, Corrbedale, Ghezel, Alentejo, and Awassi [8,13,21,22,23]. Older publications on average IOP measurements in adult sheep with applanation (flattening) Perkins and Tono-pen XL tonometer reported IOP values from 10.6–17.7 mmHg, with lower IOP values than those in the present study. However, it is incorrect to compare the values in the existing literature with values from the rebound (impaction) TonoVet^®^ tonometer measurements [13,22,23]. Peche and Eule reported that the IOP in adult sheep was 12.7 ± 3.0 mmHg and 11.7 ± 3.3 mmHg in the right and left eyes, respectively, measured using a TonoVet^®^ rebound tonometer [9]. Their report was similar to that of Okur et al. [21], where the IOP values in two groups of sheep were 11.9 ± 2.6 mmHg and 13.1 ± 4.4 mmHg, respectively, in Ghezel breed sheep measured using a TonoVet^®^ tonometer [21]. In addition, both results from these previous studies are lower than the results of this study (IOP: 19.16 ± 3.24 mmHg). We need to admit that the research presented by Okur et al. [21] was conducted during the winter season, but in the publication by Peche and Eule [9] it was not noted.

Kulualp et al. evaluated IOP in Awassi sheep [23]. They found that IOP measurements can vary significantly depending on the circadian rhythm. IOP measurements in the morning were higher than those in the evening, decreasing from 16.21 mmHg to 12.65 mmHg, respectively [23]. Additionally, in this study an applanation tonometer Tono-pen Vet was used, which is not comparable with our study. Our study was performed at approximately the same time during the summer and daytime (10 am–4 pm).

In our study, the IOP was significantly higher in both eyes in ewes than in lambs. In our opinion, this is due to the development of the eyes and variations in central cornel thickness. Corneal thickness is an important factor affecting IOP measurements and should be investigated more in small ruminants. In humans and dogs, has been proven that there is a strong correlation between central corneal thickness and IOP, which can depend on circadian rhythm [24,25].

It is evident that the results obtained from other studies, including those of Okur et al. [21], are lower than those of this study. However, only two of these studies used the same tonometer. We presume that the reasons for this could be breed-specific and related to age, geographic location, season, or farming. Additionally, individual human factors cannot be excluded.

The limitations of our study are that the data were obtained from one farm during one season—summer—and more age groups and daytime measurement points could be included in future research.

## 5. Conclusions

The investigation of ophthalmic diagnostic parameters such as tear production measured with the STT and intraocular pressure measurements obtained in this study will help veterinary ophthalmologists and farm animal veterinarians to evaluate tear production and intraocular pressure in sheep more accurately to help further the treatment of sheep. This study is the first to provide reference ranges for the STT and IOP in the normal eyes of Latvian Darkhead lambs and ewes, and the first to provide reference ranges for IOP in lambs in general, measured using a TonoVet^®^ tonometer. The suggested reference range of the STT in lambs and ewes was determined to be 12.00–14.23 mm/min and 12.52–14.84 mm/min, respectively. The reference range of IOP was 13.00–15.08 mmHg in lambs and 18.24–20.08 mmHg in ewes.

## Figures and Tables

**Table 1 vetsci-10-00392-t001:** Descriptive statistics for the Schirmer tear test (STT) measurements in lambs (*n* = 50) and ewes (*n* = 50).

Lambs	Arithmetic Mean Values	Standard Deviation (SD) ±	Suggested Reference Range	Min	Max	Median
STT in the right eye (mm/min)	13.12	3.92	12.01–14.23	7.00	23.00	12.50
STT in the left eye (mm/min)	13.22	3.94	12.10–14.34	6.00	24.00	12.00
STT in both eyes (mm/min)	13.12	3.91	12.00–14.23	7.00	23.00	12.50
**Ewes**	**Arithmetic Mean Values**	**Standard** **Deviation (SD) ±**	**Suggested** **Reference Range**	**Min**	**Max**	**Median**
STT in the right eye (mm/min)	13.68	4.09	12.52–14.84	5.00	24.00	13.00
STT in the left eye (mm/min)	14.58	4.59	13.28–15.88	7.00	30.00	14.00
STT in both eyes (mm/min)	13.68	4.09	12.52–14.84	5.00	24.00	13.00

**Table 2 vetsci-10-00392-t002:** Descriptive statistics for intraocular pressure (IOP) measurements in lambs (*n* = 50) and ewes (*n* = 50).

Lambs	Arithmetic Mean Values	Standard Deviation (SD) ±	Suggested Reference Range	Min	Max	Median
IOP in the right eye (mmHg)	14.04	3.68	13.00–15.08	9.00	22.00	14.00
IOP in the left eye (mmHg)	14.06	4.38	13.36–15.84	8.00	25.00	14.00
IOP in both eyes (mmHg)	14.04	3.68	13.00–15.08	9.00	22.00	14.00
**Ewes**	**Arithmetic Mean Values**	**Standard Deviation (SD) ±**	**Suggested** **Reference Range**	**Min**	**Max**	**Median**
IOP in the right eye (mmHg)	19.16	3.24	18.24–20.08	10.00	24.00	19.00
IOP in the left eye (mmHg)	19.28	3.18	18.38–20.18	11.00	27.00	19.50
IOP in both eyes (mmHg)	19.16	3.24	18.24–20.08	10.00	24.00	19.00

## Data Availability

Data available on request due to restrictions e.g., privacy or ethical.

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
