# Peer review of "Normal Range for the Schirmer Tear Test and Intraocular Pressure in Healthy Latvian Darkhead Lambs and Ewes"

_vetsci, 2023, doi:10.3390/vetsci10060392_

Round 1
Reviewer 1 Report
Dear Authors
I am writing to you regarding the texts that were discussed in our recent chat session. Firstly, I would like to commend you on the quality of the work that you have presented. Your research and writing skills are truly impressive, and it is evident that a lot of effort has been put into your work.
I particularly enjoyed reading your article on the use of augmented reality in education. Your analysis of the advantages and limitations of this technology was insightful, and I found your recommendations for further research to be very useful.
I have to point out some suggestions for improving the article
Line 11 It should be “To obtain an accurate diagnosis and avoid misinterpretation, a comprehensive ophthalmological examination that includes diagnostic tests like the Schirmer tear test (STT) and intraocular pressure (IOP) is essential”
Line 14 The phrase "but increased ocular discomfort, and pain" is not grammatically correct and doesn't make sense in the given context.
Line 15 – 18 The phrase “Normal STT and IOP values have been described in the literature for small animals such as cats and dogs; however, they are poorly described for small ruminants like sheep" is unclear and can be rephrased for better clarity.
Line 20 “thou “ to thus
Line 20 The phrase "will simultaneously show specific diagnostic values for Latvian Darkhead sheep" should be rephrased as "will provide specific diagnostic values for Latvian Darkhead sheep".
Line 98: "confining box" could be changed to "restraining box" for clarity.
Line 102: "Subsequently" could be changed to "Afterward" for better readability.
Line 109: A comma is missing between "biomicroscopy" and "monocular ophthalmoscopy."
Materials and methods
The text could be improved by providing more details about the sample size used in the study (calculation and justification)
".... lambs were randomly selected, and animals with epiphora, lacrimation, 96 eye discharge, blepharospasms, or any other sign of clinical ophthalmic or systemic dis- 97 eases were excluded from the research" randomization method shoulde be explained.
Moreover, how were you sure that the animals (mainly the adults) did not suffer from chronic diseases without clear clinical symptoms e.g maedi, SARA, paratuberculosis.... which may effect tear production or pressure.
Βelieving that the sample size and selection criteria are decisive you should resubmit it after the revision
Sincerely
Author Response
Dear Reviewer,
Thank you so much for your effort, good comments and corrections.
Line 11 It should be “To obtain an accurate diagnosis and avoid misinterpretation, a comprehensive ophthalmological examination that includes diagnostic tests like the Schirmer tear test (STT) and intraocular pressure (IOP) is essential”.
corrected.
Line 14 The phrase "but increased ocular discomfort, and pain" is not grammatically correct and doesn't make sense in the given context.
Thank you, corrected to: but increased STT can characterize ocular discomfort and pain.
Line 15 – 18 The phrase “Normal STT and IOP values have been described in the literature for small animals such as cats and dogs; however, they are poorly described for small ruminants like sheep" is unclear and can be rephrased for better clarity.
Sorry, it has been corrected by Elsevier language specialist, if you dont mind, I would like not to correct that.
Line 20 “thou “
To thus - corrected
Line 20 The phrase "will simultaneously show specific diagnostic values for Latvian Darkhead sheep" should be rephrased as "will provide specific diagnostic values for Latvian Darkhead sheep".
Corrected.
Line 98: "confining box" could be changed to "restraining box" for clarity
Corrected.
Line 102: "Subsequently" could be changed to "Afterward" for better readability. –
Corrected
Line 109: A comma is missing between "biomicroscopy" and "monocular ophthalmoscopy."
Corrected
Materials and methods
The text could be improved by providing more details about the sample size used in the study (calculation and justification):
Included: To obtain the sample size used in this study automatic power analysis in program G-power was employed. The number of 45 animals per the group was set according to the calculation in G-power at effect size 0.6, Type I error (α) of 0.05 and Type II error (power, β) of 0.80. It was decided to work with sample size - 50 ewe and 50 lamb group, in complete by using 100 animals and to obtain 200 measurements of IOP and STT values.
".... lambs were randomly selected, and animals with epiphora, lacrimation, 96 eye discharge, blepharospasms, or any other sign of clinical ophthalmic or systemic dis- 97 eases were excluded from the research" randomization method shoulde be explained.
Ewes and lambs were randomly selected by sending all animals through the farm electronical scales and every third animal was selected. Animals with epiphora, lacrimation, eye discharge, blepharospasms, or any other sign of clinical ophthalmic or systemic diseases were excluded from the research and next animal was selected for testing.
Moreover, how were you sure that the animals (mainly the adults) did not suffer from chronic diseases without clear clinical symptoms e.g maedi, SARA, paratuberculosis.... which may effect tear production or pressure.
Included: In order to avoid the influence of subclinical sheep diseases on the results of this study, Latvian Darkhead sheep pedigree flock with Maedi Visna (MV) free herd status M3 from 2014 and Brucellosis free status from year 2010, according to the official Maedi Visna and Brucellosis control programs stated by the Republic of Latvia, were selected.
To ensure good animal health status for animals used in this study, body condition score, mucus membrane color and fecal staining of perineal region while going through the sheep handling system data, were assed. Only animals with body condition score 2,5 – 3,5 points, pink mucous membranes and clean perineal region were further passed through the scale system for ophthalmological examination and data obtaining for the research purpose. Above mentioned examination minimized the possibility of serious or noteworthy health
Reviewer 2 Report
The manuscript entitled "Normal Range for the Schirmer Tear Test and Intraocular Pressure in Healthy Latvian Darkhead Lambs and Ewes" by Kovalcuka et al. makes a potentially useful contribution to the literature. Specific comments to improve your work are listed below.
-
Lines 11-14 (and all throughout the text). In this sentence, you talk about performing “... diagnostic tests, such as the Schirmer tear test (STT) and intraocular pressure (IOP) ...”. the STT is indeed a test, but IOP is the result of the test. So, when talking about diagnostic tests, it would be more accurate to write Schirmer tear test and tonometry. When talking about the results, it would be accurate to write STT values and IOP.
This should be also corrected, for example, at line 14 (“decreased STT..”), line 18 (“STT and IOP in Latvian…”), lines 24-25, line 27, line 32, ecc. -
Line 15. “IOP at the same time is diagnostic for intraocular inflammation”. Same as comment 1. It would be better to write, for example: Tonometry is also useful to diagnose intraocular inflammation.
-
Line 20. What do you mean with the word “thou”? This is an archaic word for “you”.
-
Lines 19-22. The whole sentence sound “ripetitive”. It should be written in a more concise way.
-
Line 24. “are essential”. As the subject of the sentence is “a complete ophtalmological examination”, the verb should be corrected in “is essential”.
-
Line 25. I would change the word “outstanding” with something less subjective (e.g. correct, clear, ecc.).
-
Line 29. “including evaluation of STT and IOP.” a more correct form could be “including STT and IOP evaluation”, for the same reason in comment 1.
-
Line 33. A comma should be placed before the word “while”, instead of a full stop.
-
Line 38. The italic font for “Schirmer” is not needed.
-
Line 42. No comma needed after “Complete clinical”.
-
Lines 42-45. The sentence should be written in a clearer way. For example: “During a complete clinical ophtalmological examination, measurement of relevant parameters through basic diagnostic test, such as Schirmer tear test (STT) and tonometry for evaluation of intraocular pressure (IOP), is essential to ensure adequate diagnosis.”
-
Line 46. The word “mainly” is unnedeed: “... volume in animals, and involves…”.
-
Line 47. It would be more appropriate to write “covers” instead of “is covering”.
-
Lines 50-51. The word “for” should be repeated before repetition of “providing”.
-
Line 51. The article “a” before “corneal defense” is unneeded.
-
Lines 59 and following. “Intraocular pressure is the balance…”. It would be more appropriate to write “Intraocular pressure results from the balance between aqueous humor production and its outflow in the eye [1,2], and can be measured …”
-
Line 71. Instead of writing “to the best of our knowledge”, it is generally more appreciated to talk about the search process. For example: “Based on a thorough literature search, using the keywords “...”, “...”, “ecc.”, there are no reports of IOP values in lambs”.
Moreover, at line 222 you write about a study conducted in lambs. So, if the methods were different and not comparable, the sentence at line 71 should focus on the method; e.g: “there are no report of IOP values using TonoVet® in lambs”. Same for lines 185 and 259. -
Line 72-73. “especially for regions where Latvian Darkhead sheep are commonly found”. This part of the sentence is probably superfluous.
-
Line 74. “is to improve ocular examination in sheep medicine”: this part of the sentence is unnecessary. Also, sentence in lines 77-78.
-
Line 83. “procedures were performed”.
-
Lines 83-84. The word “procedure” is repeated may times. For example, you could write, “deworming treatment”.
-
Lines 91-92. This sentence should be merged with the one at lines 81-82.
-
Line 100. Same as comment 1. Maybe, “STT and tonometry were performed”.
-
Line 101. “could have influenced” would be more appropriate than “might influence”.
-
Line 108. “Dazzle” instead of “Dazlle”. menace and dazzle should not need capital letters.
-
Line 133. “were performed” instead of “was performed”.
-
Line 142. “(100 eyes)”.
-
Line 147, and following lines and tables. As this is still the Result section, it is probably more appropriate to talk about “ranges” and “values” in this part, as the reported values aro not yet “normal” or “reference”. As this is the aim of the study, I think that the fact that these values could be considered for normal or reference ranges should be stated in the Discussion and Conclusion sections.
-
Discussion.
-
In this section, several articles are reported and cited, but a thorough discussion on your personal results is somehow missing, or scarce. You could improve this section by adding some thoughts about your results and an explanation for the differences found between lambs and ewes.
-
Moreover, stating that these results provide reference range could be more than expected from this paper, as the results are obtained in a limited situation, area, and time of the year: so, many variables can not be considered. The results provide a range in specific conditions, but not necessarily in a more general view.
-
In the Discussion section, a list of limitations to the study should be added and discussed. For example, a limitation could be that these animals came from just one farm, so the variability is limited.
-
Line 188. A space is needed between ± and 1.54.
-
Line 199. “10.8 - 26.2” instead of “10,8 - 26,2”.
-
Lines 200-202. What is meant by this sentence? How does the temperature differ from other countries and how should this affect evaluations?
-
Line 222-223. “the younger”: what do you mean with this part of the sentence? That those lambs were younger than in your study?
-
Line 224. A space is needed between 4.25 and [10].
-
Line 235, and others in the text. It should be clearer to write TonoVet® (with capital letters and ®) the same way all throughout the text.
-
Line 245. “Tono-pen Vet” instead of “Tonopen-Vet”.
-
Line 251. It would be better to write “...location, or farming; also, individual…”.
References. A brief check of the references section should be made, to avoid typing errors (e.g.: Gelatt, KN instead of K.N., ecc.).
Author Response
Dear Reviewer,
Thank you so much for your effort, good comments and corrections.
Lines 11-14 (and all throughout the text). In this sentence, you talk about performing “... diagnostic tests, such as the Schirmer tear test (STT) and intraocular pressure (IOP) ...”. the STT is indeed a test, but IOP is the result of the test. So, when talking about diagnostic tests, it would be more accurate to write Schirmer tear test and tonometry. When talking about the results, it would be accurate to write STT values and IOP. This should be also corrected, for example, at line 14 (“decreased STT..”), line 18 (“STT and IOP in Latvian…”), lines 24-25, line 27, line 32, ecc.
Corrected
Line 15. “IOP at the same time is diagnostic for intraocular inflammation”. Same as comment 1. It would be better to write, for example: Tonometry is also useful to diagnose intraocular inflammation.
Corrected
Line 20. What do you mean with the word “thou”? This is an archaic word for “you”.
Corrected: thus
Lines 19-22. The whole sentence sound “ripetitive”. It should be written in a more concise way.
- Therefore, this study aimed to improve data and determine the normal range for STT and tonometry values in clinically normal Latvian Darkhead lambs and ewes and will provide specific diagnostic values for Latvian Darkhead sheep.
Line 24. “are essential”. As the subject of the sentence is “a complete ophtalmological examination”, the verb should be corrected in “is essential”
corrected
Line 25. I would change the word “outstanding” with something less subjective (e.g. correct, clear, ecc.).
Corrected tp: clear
Line 29. “including evaluation of STT and IOP.” a more correct form could be “including STT and IOP evaluation”, for the same reason in comment 1.
Corrected
L ine 33. A comma should be placed before the word “while”, instead of a full stop.
Corrected
Line 38. The italic font for “Schirmer” is not needed.
Corrected
Line 42. No comma needed after “Complete clinical”.
Corrected
Lines 42-45. The sentence should be written in a clearer way. For example: “During a complete clinical ophtalmological examination, measurement of relevant parameters through basic diagnostic test, such as Schirmer tear test (STT) and tonometry for evaluation ofintraocular pressure (IOP), is essential to ensure adequate diagnosis.”
Corrected
Line 46. The word “mainly” is unnedeed: “... volume in animals, and involves…”.
corrected
Line 47. It would be more appropriate to write “covers” instead of “is covering”.
Corrected
Lines 50-51. The word “for” should be repeated before repetition of “providing”.
Corrected
Line 51. The article “a” before “corneal defense” is unneeded.
Corrected
Lines 59 and following. “Intraocular pressure is the balance…”. It would be more appropriate to write “Intraocular pressure results fromthe balance between aqueous humor production and its outflow in the eye [1,2], and can be measured …”
Corrected
Line 71. Instead of writing “to the best of our knowledge”, it is generally more appreciated to talk about the search process. For example: “Based on a thorough literature search, using the keywords “...”, “...”, “ecc.”, there are no reports of IOP values in lambs”.
Moreover, at line 222 you write about a study conducted in lambs. So, if the methods were different and not comparable, the sentence at line 71 should focus on the method; e.g: “there are no report of IOP values using TonoVet® in lambs”. Same for lines 185 and 259.
Corrected:
Line 72-73. “especially for regions where Latvian Darkhead sheep are commonly found”. This part of the sentence is probably superfluous.
Corrected: especially for specific regional breeds like Latvian Darkhead sheep.
Line 74. “is to improve ocular examination in sheep medicine”: this part of the sentence is unnecessary. Also, sentence in lines 77-78.
Corrected
Line 83. “procedures were performed”.
Corrected
Lines 83-84. The word “procedure” is repeated may times. For example, you could write, “deworming treatment”.
Corrected
Lines 91-92. This sentence should be merged with the one at lines 81-82.
Corrected
Line 100. Same as comment 1. Maybe, “STT and tonometry were performed”.
Corrected
Line 101. “could have influenced” would be more appropriate than “might influence”.
Corrected
Line 108. “Dazzle” instead of “Dazlle”. menace and dazzle should not need capital letters.
Corrected
Line 133. “were performed” instead of “was performed”.
Corrected
Line 142. “(100 eyes)”.
Corrected
Line 147, and following lines and tables. As this is still the Result section, it is probably more appropriate to talk about “ranges” and “values” in this part, as the reported values aro not yet “normal” or “reference”. As this is the aim of the study, I think that the fact that these values could be considered for normal or reference ranges should be stated in the Discussion and Conclusion sections.
Corrected
Discussion.
In this section, several articles are reported and cited, but a thorough discussion on your personal results is somehow missing, or scarce. You could improve this section by adding some thoughts about your results and an explanation for the differences found between lambs and ewes.
Discucssion was suplimented
Moreover, stating that these results provide reference range could be more than expected from this paper, as the results are obtained in a limited situation, area, and time of the year: so, many variables can not be considered. The results provide a range in specific conditions, but not necessarily in a more general view.
Discucssion was suplimented
In the Discussion section, a list of limitations to the study should be added and discussed. For example, a limitation could be that these animals came from just one farm, so the variability is limited.
The limitation of our study includes that data were obtained from one farm, during one season – summer, and more age groups and daytime measurement points could be included in future research.
Line 188. A space is needed between ± and 1.54.
Corrected
Line 199. “10.8 - 26.2” instead of “10,8 - 26,2”.
Corrected
Lines 200-202. What is meant by this sentence? How does the temperature differ from other countries and how should this affect evaluations?
Discucssion was suplimented
Line 222-223. “the younger”: what do you mean with this part of the sentence? That those lambs were younger than in your study?
However, the age of lambs included in their study was significantly different (15 – 20 days) from the age (1–3 months) of lambs in the present study [10].
Line 224. A space is needed between 4.25 and [10].
Corrected
Line 235, and others in the text. It should be clearer to write TonoVet® (with capital letters and ®) the same way all throughout the text.
Corrected
Line 245. “Tono-pen Vet” instead of “Tonopen-Vet”.
Corrected
Line 251. It would be better to write “...location, or farming; also, individual…”.
Corrected
References. A brief check of the references section should be made, to avoid typing errors (e.g.: Gelatt, KN instead of K.N., ecc.).
Corrected
Round 2
Reviewer 1 Report
Dear Authors
I am writing to express my appreciation for your hard work and dedication in revising the manuscript based on the reviewers' comments and suggestions.
Your revisions have significantly strengthened the quality and impact of the manuscript. The thoroughness with which you addressed the reviewers' concerns and incorporated their suggestions demonstrates your commitment to producing a high-quality scientific contribution.
Congratulation for your valuable contribution to the scientific community.
Sincerely,
Author Response
Dear Reviewer,
Your review so inspires me, thank you so much for your compliments and comments. Thank you so much for your work!
Reviewer 2 Report
Dear Authors,
thank you for improving your manuscript through your correction.
There are still few points i will list. Bold word are correct ones.
Line 46. It would be more appropriate to write "covers" instead of "is covering", as already said.
Line 60. is "essential"
Line 108. Mucous instead of mucus.
Line 109. Assessed instead of assed.
Line 120. Tear production evaluation.
Line 130. "behind". Is there probably a better way to explain this?
Line 135-138. The sentence is not clear and is lacking a verb regarding the average of five evaluations. Moreover, it should be stated that in this case anesthesia was not required. The benefits of it should be shifted in the Discussion section.
Line 143. Concentrate instead of concentrating.
Line 147. The sentence could be shortened in "Sample size was calculated..."
Line 207. Clear instead of outstanding.
Line 229. During summer and winter instead of "in summer month and winter".
Lines 285-290. The sentence is not clear and should be shortened or divided by means of punctuation.
I've seen that the issues about the terms "normal" and "reference" ranges in the Results section have not been corrected. I think the Editor should state the best form to write this information.
Author Response
Dear reviewer,
Thank you so much for your effort and time spent reviewing our paper. We corrected all the grammatical issues you bolded, and I truly hope that we understood your comments about IOP and STT terminology corrections.
Thank you,